# The Influence of Multilayer, "Sandwich" Package on the Freshness of Bread after 72 h Storage

**Małgorzata Mizielińska \*, Urszula Kowalska, Alicja Tarnowiecka-Kuca, Paulina Dzięcioł, Katarzyna Kozłowska and Artur Bartkowiak**

Center of Bioimmobilisation and Innovative Packaging Materials, Faculty of Food Sciences and Fisheries, West Pomeranian University of Technology Szczecin, Janickiego 35, 71-270 Szczecin, Poland; urszula.kowalska@zut.edu.pl (U.K.); alicja.tarnowiecka-kuca@zut.edu.pl (A.T.-K.); paulina.dzieciol@zut.edu.pl (P.D.); katarzyna.kozlowska@zut.edu.pl (K.K.); artur.bartkowiak@zut.edu.pl (A.B.)
\* Correspondence: malgorzata.mizielinska@zut.edu.pl; Tel.: +48-91-449-6132

**Abstract:** The goal of this research was to evaluate a polymeric system, including biopolymers, as a multi-component material coating on paperboard for bread. The main aim of the research was to create a humidity-controlling packaging material. This means that the packaging material should contain filler which will absorb water or water vapour from the bread. The ideal packaging should have high barrier qualities against water vapour, enabling the possible release of water from the product (to maintain proper humidity inside the packaging). The preliminary storage tests made of the bread confirmed that the freshness of a product kept in a climatic chamber in RH = 70% was the highest. To summarise, the obtained packaging should maintain the required humidity (in the case of bread, the optimal humidity is 70%) within the packaging to keep the bread fresh after more than 72 h of storage. A "sandwich" form of (multilayer) packaging was indicated as a solution to this problem. The main objective of this packaging was to obtain two paper layers and one starch layer to increase water absorption from the bread. It was also important to obtain a thin, external hydrophobic layer to decrease the water vapour transmission rate of the packaging (WVTR). A number of "sandwich" packaging types were prepared, consisting of two sheets of paper with an external aquaseal coating and an internal starch coating (with a NaCl filler). The covered "sandwich" papers were then used to create packaging that could be used for the bread storage tests. The study results confirmed that the bread stored for 72 h in the "sandwich" packaging was found to be fresher than the same product stored in commercial paper packaging.

**Keywords:** bread; water vapour; water absorption; multi-component materials' coatings; paperboard; "sandwich" packaging; humidity-controlling packaging; packaging material

## 1. Introduction

Due to its high-quality nutritional properties and sensorial characteristics, bread is a staple part of our diet. The shelf life of bread is relatively short, limited as it goes stale, the main responsible factor for economic loss to both the baking industry and the consumer [1]. Normally, the shelf life of bread without the use of any preservation methods is around 3–4 days. Staling is one of the most important factors limiting its shelf life, and it is a complicated phenomenon, and the exact mechanism is not completely understood. It is suggested that processes, such as cross-linking between starch and protein, partial drying, glassy-rubbery change and moisture transformation from the crumb to the crust of the bread are involved in the staling process. Generally, staling is divided to crust staling and crumb staling. Crust staling is attributed to the migration of moisture from crumb to crust, while crumb staling is related to a physicochemical alteration in starch [2–4]. Starch retrogradation and water loss

are effects that share the same intensity at the increase of bread firming. Water migration results in an equilibration of the water content between the crust and the crumb at a macroscopic scale and in the redistribution of moisture between the components at a microscopic scale. This equilibrium is unstable and may drift towards crust softening and crumb drying [5,6]. The short shelf life of bread and its staling depends on various factors: the bread-making process, baking conditions and storage conditions, such as room temperature, relative humidity and storage with or without crust [4]. Packaging is the last step of production, and food technologists have to select the most suitable type of packaging to ensure the longest shelf life. A successful product is based equally on the intrinsic quality of a product and the effectiveness of the packaging in preserving this quality. Several studies have offered evidence on the efficacy of packaging in maintaining the quality characteristics of bread and slowing down moisture loss by using suitable materials [7,8]. Active packaging is a promising solution in food production, such as bread shelf-life extension. It is a novel system that changes packaging conditions, thereby extending the shelf life of the product and improving safety, as well as sensory properties, while maintaining food quality [8,9]. Paper packaging is widely used in various industries. In addition to its effective cost and high application flexibility, paper is also considered the most environmentally friendly material for packaging, in view of its natural sourcing and easy recyclability [8]. Several studies have shown active cellulose-based papers modified with $TiO_2$, $Ag$–$TiO_2$ and $Ag$–$TiO_2$–$Z$ nanocomposites or cellulose acetate films incorporated with sodium propionate or using ethanol emitter, oxygen absorber, water absorber and oxygen and/or water scavenger packages that can be used for bread packaging [1,9]. Paper packaging is biodegradable and cheap, but it is characterised by high water vapour permeability; as it is released from the packed bread, it escapes to the outside of the packaging, which in turn contributes to bread staling. A solution to this problem could be the modification of paper packaging by the use of a coating.

The aim of the experiments was to obtain multi-layer paper packaging with an inner and outer coating. The purpose of the internal starch/NaCl-based coating was to absorb the water released by the packed bread. Both starch and NaCl are good water absorbers. The purpose of the external coating (based on Aquaseal 2258—water-based polyethylene coatings) was to increase the water vapour barrier. This multi-layer packaging would be able to hold the moisture inside the packaging and prevent the bread from becoming stale.

## 2. Materials and Methods

### 2.1. Materials

Fresh wheat rye bread (Lidl, Neckarsulm, Poland) was bought and transported (in cardboard boxes) to the Centre of Bioimmobilisation and Innovative Packaging Materials (CBIMO).

Kraft Paper (grammage 35 $g/m^2$) (Nordic Paper, Greåker, Norway) was used in this research. Starch C Film 07311 (Cargill, Warsaw, Poland) and Aquaseal 2258 (Paramelt, Heerhugowaard, The Netherlands) was used as a coating carrier, with Sodium chloride (Sigma-Aldrich, Inc. Merck KGaA, Darmstadt, Germany) used as an absorber. To verify the salt and sugar content in the bread, chemicals such as: $AgNO_3$, $K_2CrO_4$, $CuSO4$, $NaCO3$ and $C_6H_5Na_3O_7$ (Sigma-Aldrich, Inc. Merck KGaA, Darmstadt, Germany) were used.

### 2.2. Modification of Paper Package

25 g of starch was introduced into 64.6 mL of water. The mixture was mixed for 45 min at 90° using a mechanical stirrer (Ika). After starch gelatinisation, the starch gruel was cooled down. As a next step, 10.4 g of NaCl was introduced into 89.6 g of the starch gruel.

A sheet of the paper was applied with the starch coating containing NaCl using Unicoater 409 (Erichsen, Hemer, Germany) and then covered with a second sheet of paper. The covering was performed at a temperature of 25 °C with a 24 μm diameter roller. The coatings were dried for 5 min at a temperature of 70 °C. A 37 ± 5 g internal layer of coating per 1 $m^2$ of paper was obtained. To prepare

an external coating layer (to increase the water vapour barrier), the paper was covered with Aquaseal 2258 using Unicoater 409 (Erichsen, Hemer, Germany) with a 24 μm diameter roller. The coating was then dried for 5 min at 70 °C.

To create bags, paper sheets and multilayer paper sheets (Figure 1) were covered with a 1 cm wide layer of Aquaseal 2258. The sheets were then joined together and fused using a welder (HSE-3, RDM Test Equipment, Hertfordshire, UK) in normal air conditions. The welding parameters were: temperature: 70 °C, pressure: 3 kN and time: 1 s.

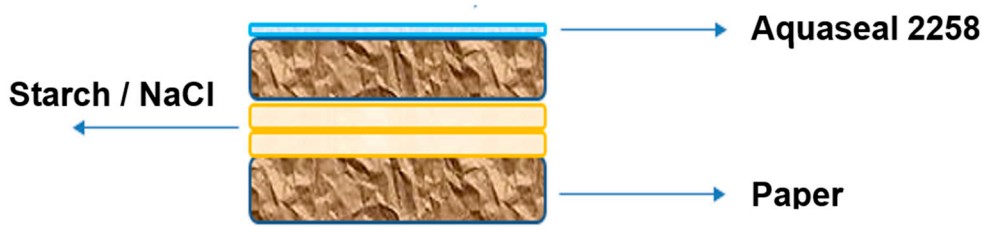

**Figure 1.** Multilayer package (sandwich packaging).

### 2.3. Packaging and Storage

Rye wheat bread (Lidl) was cut into 2 pieces. The bread portions were packed into paper and modified paper (sandwich packaging) welded bags. The samples were aseptically introduced into:

a   Paper bags (control samples).

b   Sandwich packaging bags covered with an Aquaseal 2258 as an external coating and with starch/NaCl coating as an internal coating.

The bread samples were not allowed to be in contact with the coatings.

Next, the bags were closed using a welder (HSE-3, RDM Test Equipment, Hertfordshire, Great Britain) in normal air conditions. The welding parameters were: temperature: 70 °C, pressureL 3 kN and time: 1 s.

The bags containing bread samples were then stored at 25 °C in 50% relative humidity (RH). The samples were examined after 24, 48 and 72 h storage.

### 2.4. Scanning Electron Microscopy (SEM)

An analysis (SEM) was performed using a Vega 3 LMU microscope (Tescan, Brno-Kohoutovice, Czech Republic) scanning electron microscope (SEM). The tests were necessary to determine the modified packaging material layers. An analysis was performed at room temperature with tungsten filament, and an accelerating voltage of 20 kV was used to capture SEM images for both the non-stored and stored packages. All specimens were viewed from above.

### 2.5. Relative Humidity (RH) Analysis

To determine the relative humidity (RH) inside the paper packaging (paper and sandwich packaging) containing bread, loggers to maximise integration were introduced into each package separately. The RH was measured after 24, 48 and 72 h of storage.

### 2.6. Mechanical Analysis

A textural analysis of the bread was carried out according to the PN-ISO 11,036:1999 standard: "Sensory analysis. Methodology. Texture profiling" [10]. Additionally, the hardness of bread crust and crumbs was analysed. The tests were carried out using Zwick/Roell Z 2.5 (Polish Company, Wrocław, Poland).

### 2.7. Humidity and Weight Loss of Bread

A humidity and weight loss analysis of the bread was carried out according to the PN-A-74108 standard [11].

### 2.8. Sugars and NaCl Content in Bread

Sugar content in the bread, before and after storage, was carried out using the Loof-Schoorl method. The salt content (chloride ions concentration) was determined by titration (Mohr's method). All tests were performed according to the PN-A-74108 standard [11]. The tests were carried out using Zwick/Roell Z 2.5.

### 2.9. Sensory Analysis

Crumb and crust samples for sensory tests were taken according to the PN-A-74 104:1986 (PN-86/A 74104) standard. In the case of bread crust, the colour and appearance of the surface, elasticity, crunchiness, thickness, taste and aroma were analysed. In the case of the crumbs, the colour, porosity, elasticity, taste and smell were analysed. All tests were performed according to the PN-A-74108 standard.

### 2.10. Statistical Analysis

Statistical significance was determined using an analysis of variance (two-way ANOVA). The values were considered as significantly different when $p < 0.05$. All analyses were performed with GraphPad Prism 8 (GraphPad Software, San Diego, CA, USA).

## 3. Results

Scanning electron microscopy (SEM) is the most widely applied technique to characterise packaging material layer size and morphology. Figure 2 shows a SEM image of an Aquaseal coating on the surface of the modified packaging material. The SEM image revealed that the NaCl/starch (SNaCl) layer was homogeneously distributed throughout the paper surfaces for both packaging materials before and after storage. It was observed that after storage, the package layers (except the Aquaseal layer) were wider than the layer of the package before storage. Figure 2b shows "swollen and bloated" layers. This may suggest that the layers absorbed water.

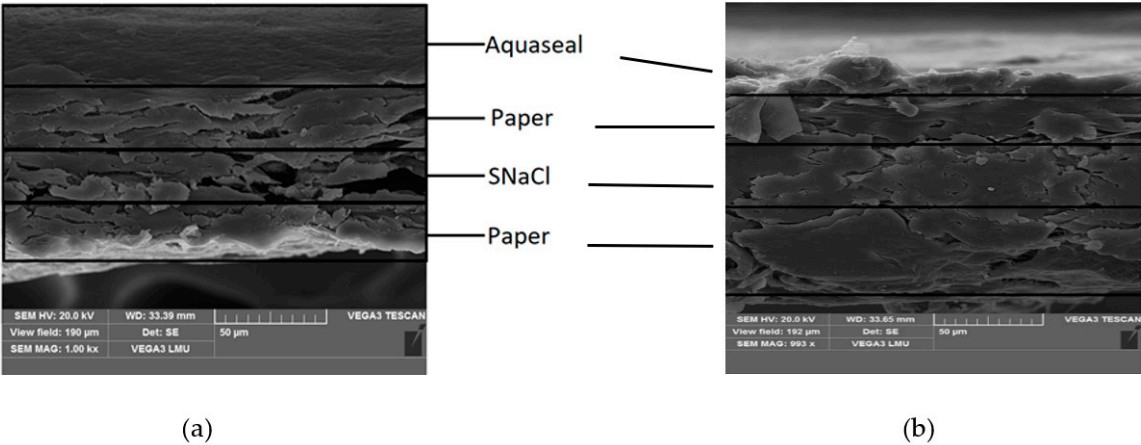

(a)    (b)

**Figure 2.** (**a**) Scanning electron microscope (SEM) image of the sandwich package before storage, (**b**) SEM image of the sandwich package after storage.

The modification of the packaging by the addition of NaCl as a water absorber and an Aquaseal coating improved the quality of the packaging. It was observed that RH inside the paper packaging containing the bread was found to be 68%, while inside the sandwiched paper it was 72% (Figure 3).

As highlighted in Figure 3, the RH inside the paper packaging increased to 74% after 24 h of storage, which was caused by water vapour released by the bread. This showed that after 48 and 72 h of storage, the RH decreased to 69% and 66%. The water vapour transmission of the paper packaging was very high and would explain why the water vapour was released by the packaging. Low RH inside the paper packaging could lead to the bread going stale and demonstrate a weak sensory quality. Alternatively, RH inside the sandwich packaging containing the bread increased to 79% after being stored for 24 h. This did not change after being stored for 48 h. After 72 h, the RH only decreased slightly (to 78%), proving that the water vapour released by the bread was absorbed by the NaCl/starch layer from the modified packaging. This was quite dissimilar to the paper packaging, as the packaging covered with the Aquaseal coating did not contribute to the release of water vapour outside it because this coating has a higher water vapour barrier than the paper. The water vapour was released from the NaCl/starch layer to the atmosphere inside the packaging, preventing the bread from going stale.

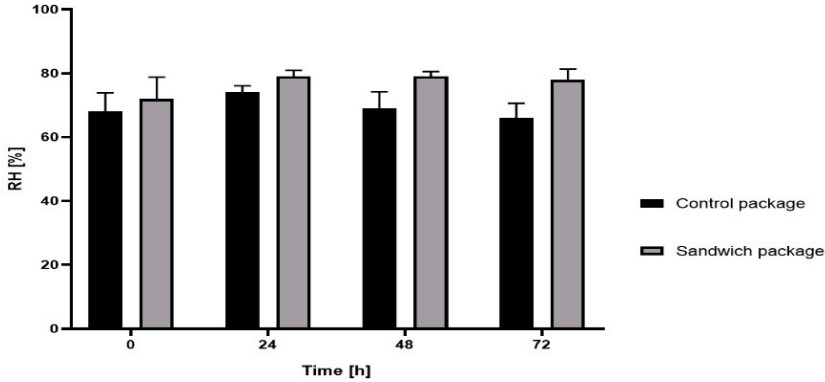

**Figure 3.** Relative humidity (RH) inside the paper and sandwich packaging after storage.

The results of the study demonstrated that bread crumb hardness increased after 72 h of storage in the control packaging and in the sandwich packaging. An increase in this parameter was not significant. Any influence of sandwich packaging on bread crumb hardness was not observed (Figure 4a). The differences between these parameter values were not found to be significant, and this was confirmed by statistical analysis ($p > 0.05$). Bread crust fillet hardness increased after being stored for 8 h in paper packaging and in sandwich packaging, with this parameter decreasing after 24 h. A modification of the packaging with NaCl as a water absorber and an Aquaseal coating as a water vapour barrier caused a smaller increase in bread crust hardness after 48 and 72 h of storage (Figure 4a). The differences between hardness values were found to be significant, and this was confirmed by statistical analysis ($p < 0.05$). As observed in Figure 4a, after 72 h of storage, bread crust hardness stored in the sandwich packaging was higher than bread crust hardness before storage, though the difference was slight. On the other hand, bread crust hardness stored in paper packaging was significantly ($p < 0.05$) higher than the bread crust hardness before 72 h of storage. The results confirmed that sandwich packaging had a clear influence on the improvement in bread freshness after 72 h of storage. An analysis of the hardness parameter suggests that the sandwich packaging could increase bread storage time. In the case of cohesiveness, it was observed that the average parameter value for breadcrumbs stored in the control packaging and in the sandwich packaging decreased after 24 h of storage (Figure 4b). After 48 h of storage, changes in breadcrumb cohesiveness was not observed. The average cohesiveness value of breadcrumbs stored in paper packaging, measured for the same samples, but stored for 72 h, increased, as well as the average cohesiveness value of the breadcrumbs stored in sandwich packaging. These changes were significant and confirmed by a two-way ANOVA test ($p < 0.05$). A decrease in this parameter was observed for samples stored in sandwich packaging and was greater than for the samples stored in paper packaging. The differences between the cohesiveness values of bread crust stored for 48 h (in both kinds of packaging) were not found to be significant, again confirmed by statistical analysis ($p > 0.05$). A significant decrease in this

parameter was observed for samples stored for 72 h in paper packaging (control samples) ($p < 0.05$).
After 72 h of storage, a significant decrease in bread crust cohesiveness for the samples stored in the
sandwich packaging was not observed ($p > 0.05$).

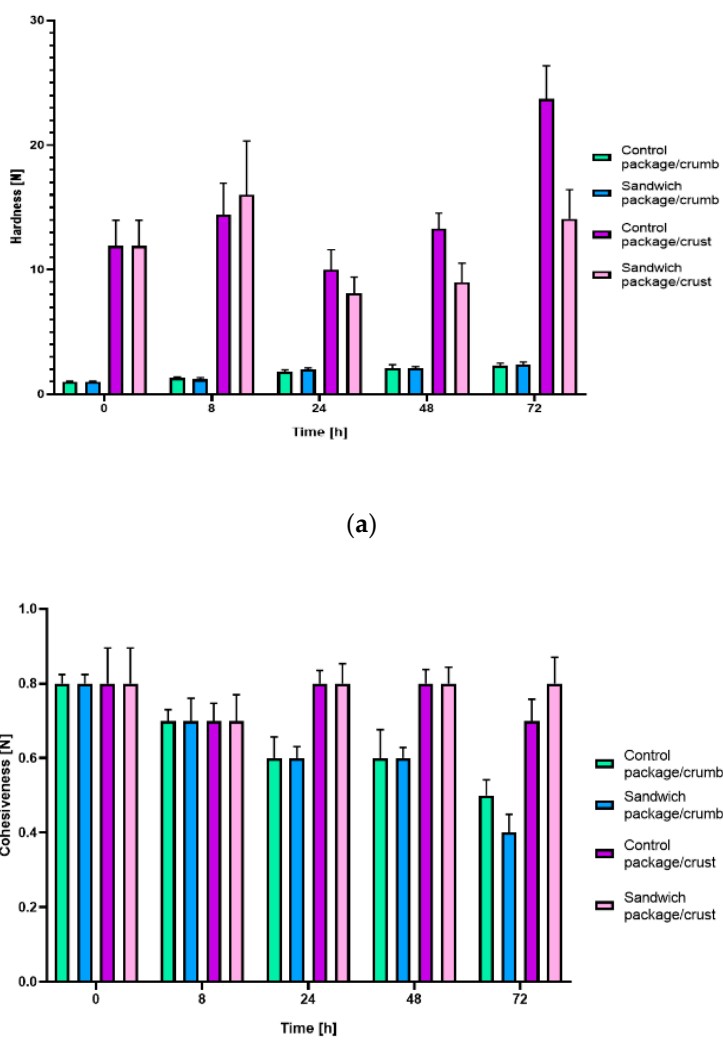

(**a**)

(**b**)

**Figure 4.** (**a**) Hardness of the crust and crumb after storage in paper and in the modified package.
(**b**) Cohesiveness of the crust and crumb after storage in paper and in the modified package.

As emphasised below (Figure 5a), the gumminess of breadcrumbs stored in the paper packaging
and in the sandwich packaging increased slightly after 72 h when compared to the ''0'' sample.
These changes were not significant ($p > 0.05$). It was demonstrated that both paper and sandwich
packaging caused a significant decrease in this parameter after 24 h of storage, again, confirmed by
statistical analysis ($p < 0.05$). After 48 h of storage, the average gumminess value of the bread crust
stored in sandwich packaging increased when compared to the gumminess value of the samples stored
for 24 h and decreased compared to the "0" sample. The opposite results were observed for samples
stored in the paper packaging, a decrease of the gumminess value was observed after 24 and after
48 h of storage. Additionally, after 72 h of storage, the gumminess of the bread crust stored in the
sandwich packaging showed a slight increase compared to the gumminess value obtained from the "0"
sample. The differences between gumminess values were not significant, which was later confirmed by
a statistical test ($p > 0.05$). Contrary to these results, a significant increase ($p < 0.05$) in the gumminess

of the bread crust stored in paper packaging was observed compared to the gumminess value obtained from the "0" sample. The results of the study demonstrated that the springiness of breadcrumbs decreased after 72 h of storage in both paper and in sandwich packaging. These changes were significant and confirmed by a two-way ANOVA test ($p < 0.05$). In the case of bread crust, after 24 h, this parameter increased, but only for samples stored in paper packaging. After 48 h, the differences between springiness values were not observed (for both kinds of packaging). A modification of the paper with an Aquaseal coating and NaCl as water absorber caused a greater increase in bread crust springiness after 72 h of storage (Figure 5b). The differences between springiness values were found to be significant, and this was confirmed by statistical analysis ($p < 0.05$).

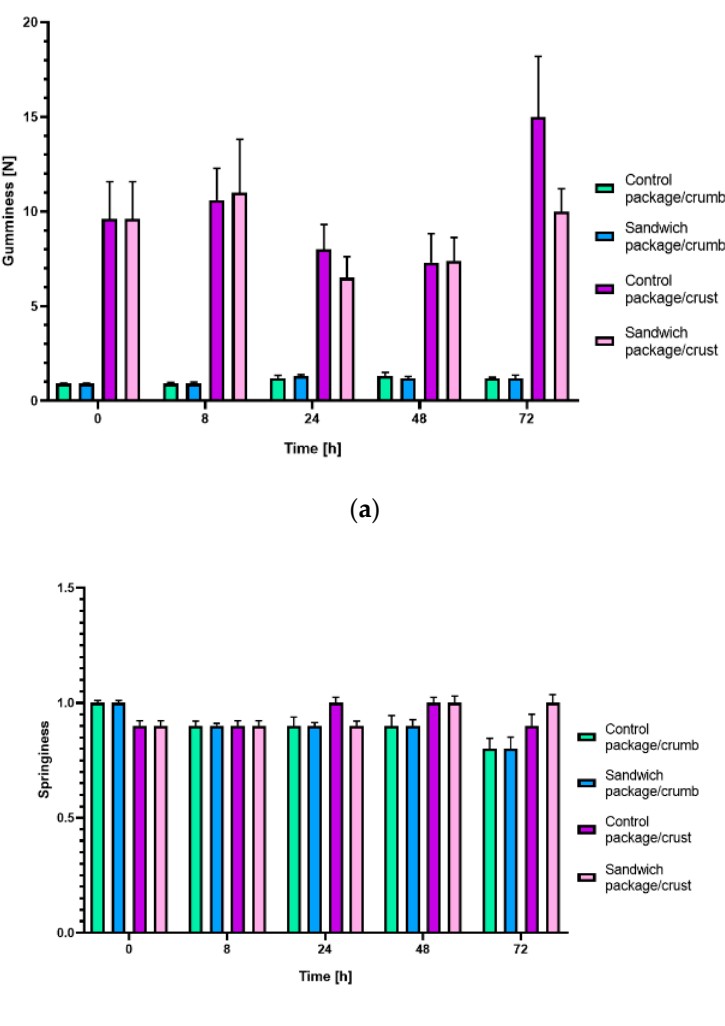

**Figure 5.** (**a**) Gumminess of the crust and crumb after storage in paper and in the modified package. (**b**) Springiness of the crust and crumb after storage in paper and in the modified package.

The results of this study demonstrated that the humidity of the breadcrumbs was 48%. The storage of bread in the paper packaging and in the sandwich paper led to a decrease in the humidity of breadcrumbs to 46% after 48 h of storage and to 43% for samples stored in paper packaging and to 44% after 72 h. It was noted that the humidity of breadcrumbs stored in sandwich packaging was higher than the humidity of the samples stored in paper packaging. The results showed that the bread crust humidity was 26%. The storage of bread in paper packaging and in the sandwich paper led to an increase in the humidity of the bread crust to 27% after 24 h of storage. After 48 h, the humidity of bread crust decreased to 22%, but only for samples stored in paper packaging. After 72 h of storage,

the humidity of bread crust decreased to 20% (for samples stored in paper packaging) and to 25% (for samples stored in sandwich packaging). It was noted that a decrease of the humidity of bread crust stored in paper packaging was significant, and this was confirmed by statistical analysis ($p < 0.05$). Paper packaging led to a decrease in the humidity of the bread crust. A humidity decrease was significantly lower for the samples stored in sandwich packaging (Figure 6).

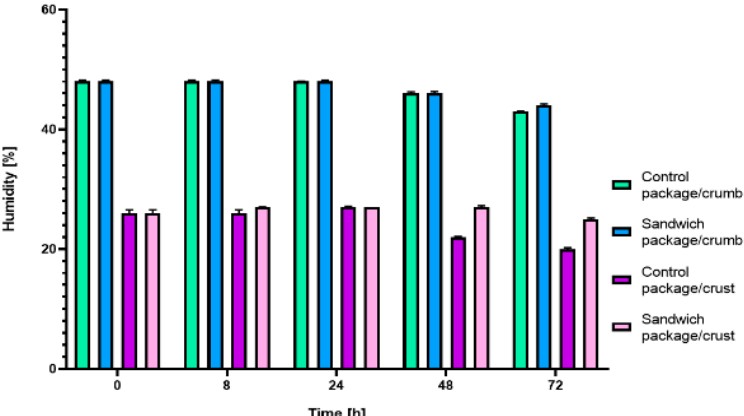

**Figure 6.** Humidity of the crust and crumb after storage in paper and in the modified packaging.

The results of the study demonstrated that the bread lost weight after 24, 48 and 72 h of storage. It was noted that the weight loss was significantly greater for the samples stored in the paper packaging than for the samples stored in the sandwich package (Figure 7).

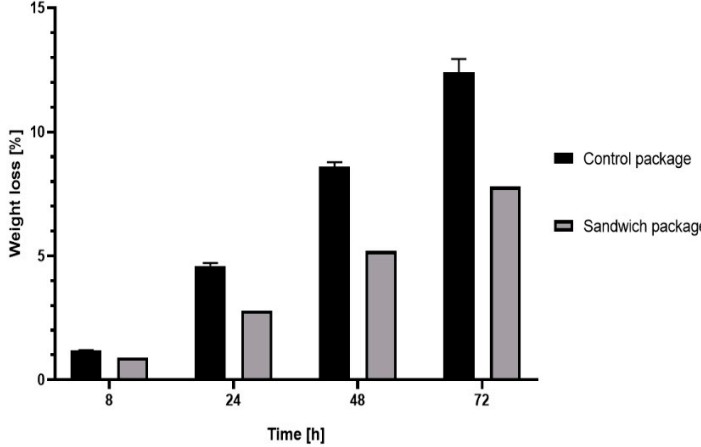

**Figure 7.** Weight loss of the bread after storage in paper and in the modified package.

It was shown in the study that the salt content of the breadcrumbs and the bread crust did not decrease during 72 h of storage (Figure 8a). It was also observed that the amount of the salt in the crust increased after 72 h of storage, but the sandwich packaging did not influence the salt content in the bread as the differences between salt content values for samples stored in sandwich packaging and in paper packaging were not significant ($p > 0.05$). The results of the study demonstrated that the sugar content of the breadcrumbs decreased slightly after 48 and 72 h of storage. The influence of the packaging modification on sugar content in the crumbs was not noted (Figure 8b) and the differences between these results were not found to be significant ($p > 0.05$). In the case of the crust, a 1% decrease in the content of sugars was observed, but only for samples which were stored in paper packaging. The sugar content did not decrease in the case of samples stored in the sandwich packaging.

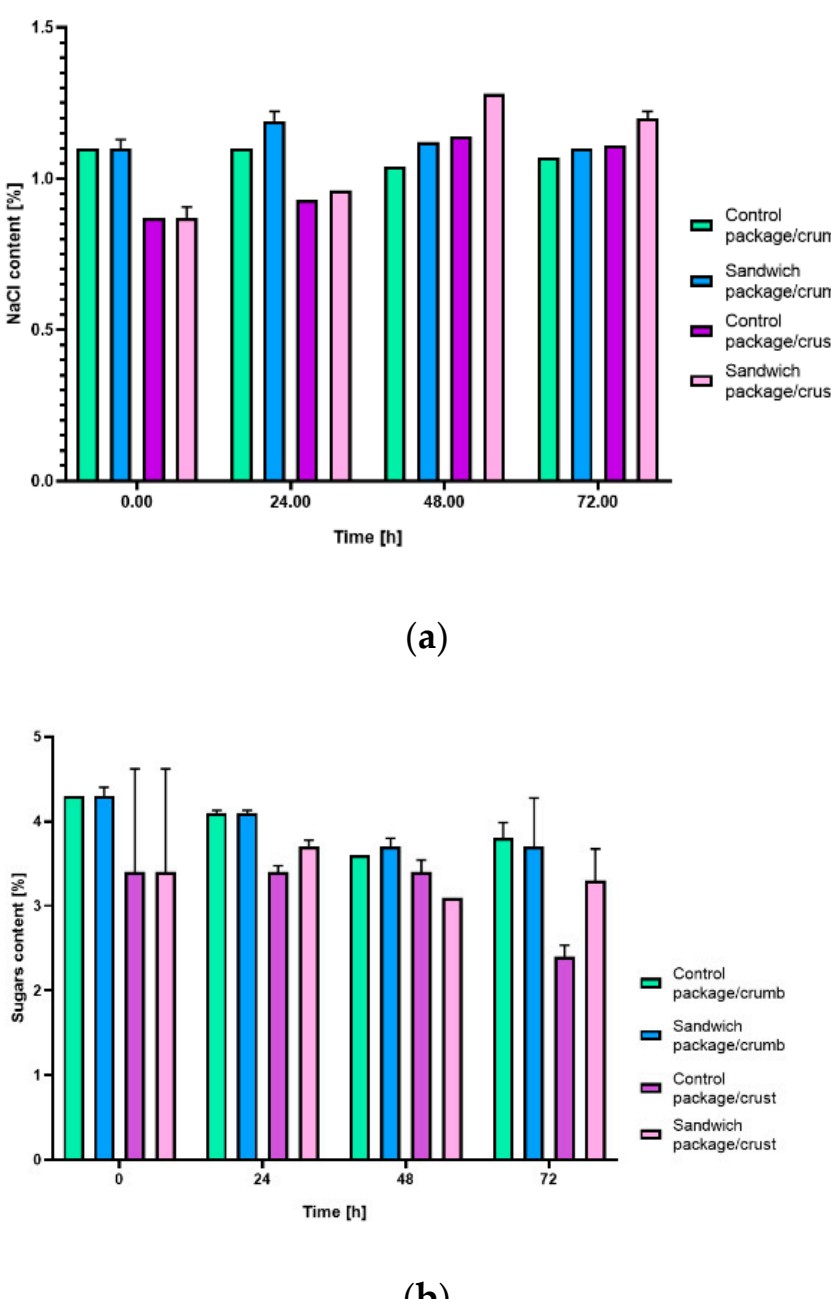

**Figure 8.** (**a**) NaCl content in the bread after storage in paper and in the modified package. (**b**) Sugar content in the bread after storage in paper and in the modified package.

Fresh bread was characterised by a high sensory quality, showing a maximum score value of 4. After 24 h of storage, the score value decreased to 2.7 for bread stored in paper packaging, while in the case of bread stored in the sandwich packaging, this only decreased to 3.6 (Figure 9). After 48 h of storage, the influence of packaging modification on the sensory quality of bread was found to be higher. According to the sensory quality evaluation (Figure 9), paper packaging gave a strong objectionable smell and taste, and the scores fell dramatically after 72 h of storage (scores value 1.1), while sandwich packaging did not markedly decrease the sensory quality of the bread (a score value of 2.5). The differences between score values for bread stored in sandwich packaging and in paper packaging were found to be significant ($p < 0.05$).

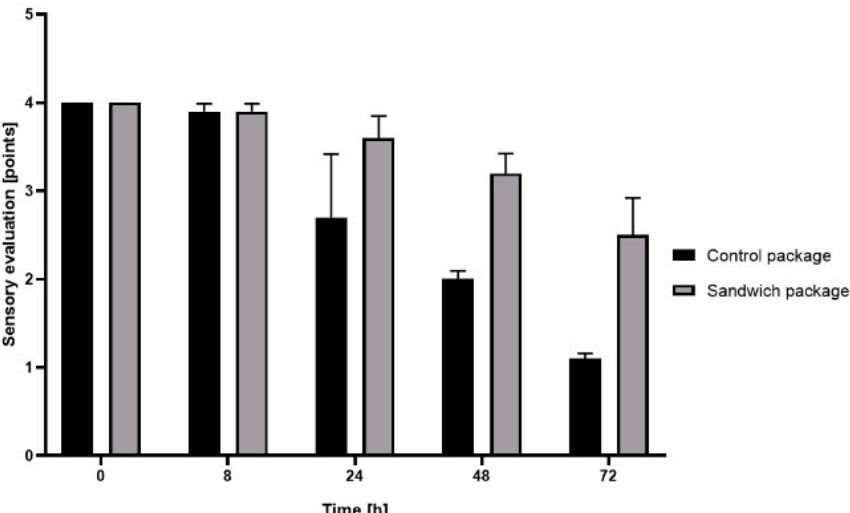

**Figure 9.** Sensory analysis of the bread after storage in the paper and the modified package.

## 4. Discussion

Modification of the packaging by the addition of NaCl as a water absorber and then covering it with the Aquaseal coating improved the quality of the packing. The results of the study demonstrated that the bread lost weight after 24, 48 and 72 h of storage. It was noted that the weight loss was significantly greater for the samples stored in paper packaging than for the samples stored in the "sandwich" packaging. It was also observed that after 72 h of storage, RH inside the paper packaging containing the bread decreased and was lower than the RH inside the "sandwich" paper (in which the RH increased). The water vapour transmission rate of the paper packaging was very high and is the reason why the water vapour was released in this case. Low RH inside the paper packaging could cause bread staling and demonstrate a weak sensory quality. Meinders et al. [12] demonstrated that initially, the bread sample may lose moisture due to the transportation of water from the surface area of the sample to the air. At the same time, moisture can also migrate from the surface area to the interior of the sample. When the surface concentration decreases below the equilibrium value, the bread sample will take up water again, and in this case, it would make sense that the moisture be inside the packaging, though it cannot be released from it. Many researchers observed moisture content loss in bread packaging in polyethylene (PE) bags during aging, even if the WVTR of the PE bags is higher than WVTR of the paper bags. Marinopoulou et al. [13] determined that a decrease in the surface layer moisture content could be attributed to the diffusion of water in the space between the polyethylene bags and the surface layer of the bread. That is, the water closer to the surface of the crumb evaporates more easily and as a result, the moisture loss in the surface layer of the crumb is higher than that of the crumb as a whole, and in the intermediate part of the crumb as well. A similar explanation was given by Le-Bail et al. [14], who found that the moisture content of breadcrumb was decreased during storage, probably due to water trapped by the polyethylene film or due to leakage into the atmosphere. A solution to this problem may be the addition of a water absorber that will absorb and release the water inside the packaging. The results of the study determined that water vapour was released by the bread and was absorbed by the NaCl/starch layer from the modified packaging. In contrast to the paper packaging, the modified packaging covered with the Aquaseal coating did not contribute to the release of water vapour outside the packaging, because this coating has a higher water vapour barrier than paper. As a next step, the water vapour was released from the NaCl/starch layer into the atmosphere inside the packaging, which prevented the bread from going stale.

As was observed in this study, after 72 h of storage, the hardness of the bread crust stored in the "sandwich" packaging was higher than the hardness of bread crust before storage, though only slightly. On the other hand, the hardness of the bread crust stored in the paper packaging was significantly

higher than the hardness of the bread crust before 72 h of storage. The results confirmed that the "sandwich" packaging had an influence on the improvement of bread freshness after 72 h of storage. An increase in hardness, dryness and crumbliness of the crumb and a fall in its resilience, along with a loss of its fragrance, were observed by Sheng et al. [6] within 24 h of storage. In the case of cohesiveness and gumminess, the differences between these parameters obtained for samples stored in "sandwich" packaging and those obtained for fresh samples (before storage) were not observed. The results proved that the modified packaging kept the bread fresh after storage and this could extend the shelf life of the product. Bosmans et al. [15] mentioned that during storage, fresh bread loses part of its desired texture associated with freshness. The crumb firms, and the crust loses its fresh bread crispiness. Results that showed that cohesiveness and gumminess changed for samples stored in paper packaging are confirmation of this thesis. Analysing hardness, gumminess and cohesiveness parameters, it could be said that the "sandwich" packaging could extend bread storage time. Springiness is a parameter that can show the springiness of the breadcrumb after being compressed once and can be defined as bread crumb elasticity. It is also known that this parameter determines the staling degree of bread [16]. The results of the study demonstrated that breadcrumb springiness decreased after 72 h of storage in both paper and sandwich packaging. Analysing the changes in the springiness of the bread, it could be said that the sandwich packaging did not improve the quality of the bread after being stored for 72 h. Similar results were obtained by Boz et al. [16], who proved that bread springiness, when wrapped up in polyethylene bags, decreased significantly after being stored for 3 days at room temperature. In comparison to the authors' results, Upasen et al. [17] also used polyethylene packaging to improve bread quality. The authors used a single polyethylene layer incorporating an oxygen scavenger, a single LDPE layer containing an oxygen absorber sachet, and three layers of PE laminated with O-nylon. The effects of the modified packaging on shelf life at atmosphere: 5, 10 and 21 vol. oxygen% in a nitrogen balance, were also analysed. The authors proved that both the scavenger and MAP systems demonstrated high potential for bread storage.

Sensory analysis performed by many authors [15] demonstrated that during storage, fresh bread loses part of the desired aroma associated with freshness, so the feeling of fresh bread disappears. Analysing the sensory quality of bread, Błaszczak et al. [18] found that during staling, porosity decreased, and crumb pores became smaller and rounder. On the other hand, Caballero et al. [19] reported that breadcrumb shrank during storage. The authors showed that storage decreased bread quality and freshness. The results of this study demonstrated that fresh bread was characterised by high sensory quality, showing a maximum score value of 4. After 72 h of storage, the score value decreased dramatically to 1.1 for bread stored in paper packaging, while for bread stored in the "sandwich" packaging, it only decreased to 2.5, respectively. The results proved that the influence of the packaging modification on the sensory quality of bread was high.

The results obtained clearly show that after 72 h of storage in "sandwich" packaging, the bread was still fresh, compared to the same product stored in paper packaging. The modified packaging improved the quality of the bread and extended its shelf life. Based on these results, it could be said that the bread could even be stored for more than 72 h. Similar results were obtained by Mihaly-Cozmuta et al. [9], who used three active cellulose-based papers containing TiO2, Ag–TiO2 and Ag–TiO2–zeolite nanocomposites (P–TiO2, P–Ag–TiO2, P–Ag–TiO2–Z) for bread packaging preparation. After the storage tests, the authors confirmed that of all of the papers investigated, those containing Ag–TiO2 and Ag–TiO2–Z proved to be the most effective in the preservation of bread stored at 4 and 20 °C, respectively. The high content of active agents in modified papers positively affected the hydrophobicity, as well as the water vapour permeability of the packaging compared to P–P (plain paper). By using modified cellulose packaging, the authors extended the shelf life of the bread up to 10 days.

Bread is a dynamic system undergoing physical, chemical and microbiological changes that limit its shelf life. Physical and chemical changes determine the loss of freshness, in terms of desirable texture and taste, and lead to the progressive firming-up of the crumb [20]. "Sandwich" packaging

improved the quality of stored bread, even if it is a complicated dynamic system. This packaging could also be used to improve the quality of other bakery products, such as cakes and biscuits, etc.

## 5. Conclusions

The modification of packaging by the addition of NaCl as a water absorber and then covering it with an Aquaseal coating improved the quality of the packaging. After analysing weight loss, humidity, sensory quality, hardness, gumminess and cohesiveness, it could be said that the modified packaging extends the storage time of bread compared to standard paper packaging.

**Author Contributions:** M.M. and U.K. conceived and designed the experiments, and analysed the data; M.M. wrote the paper; U.K. and P.D. covered paper with the coatings and prepared the packaging; A.T.-K., U.K., P.D. and K.K. performed the storage tests; A.T.-K., P.D. and K.K. prepared reagents/materials; M.M. contributed analysis tools; A.B. performed statistical analysis; M.M. and U.K. analysed the data. All authors have read and agreed to the published version of the manuscript.

**Funding:** The research work has been funded under the CORNET Programme (as the part of research project HumidWRAP CORNET/23/1/2018) by AiF and the German Federal Ministry for Economic Affairs and Energy (BMWi), Germany, by Service Public de Wallonie (SPW), and Agentschap Innoveren & Ondernemen, Belgium, and by the National Centre for Science and Development (NCBiR), Poland.

**Acknowledgments:** We would like to acknowledge the funding support, and we also wish to thank the CORNET Coordination Office and the supporting industrial partners.

**Conflicts of Interest:** The authors declare no conflict of interest.

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
