# Peer review of "The Influence of Multilayer, “Sandwich” Package on the Freshness of Bread after 72 h Storage"

_coatings, doi:10.3390/coatings10121175_

Round 1

Reviewer 1 Report

This paper considers the properties of a novel multi-layered humidity-controlling packaging material for bread packaging. The aim of the study is correctly formulated and a number of standard analytical methods have been applied for comparison of the control paper coating and a sandwich-type one with the slight, but statistically reliable differences observed. The conclusions are quite obvious and supported by the data obtained.

There are some suggestions and recommendations to the authors:

  • The full name of the abbreviation “WVTR” should be included in the Abstract.
  • In order to improve the quality of the research and to address the physical and chemical mechanisms of the bread staling process it is recommended to use NMR relaxometry for the water content estimation both in bread and in the packaging material, and also to perform a number of discrete measurements at different storage times. Simple RH% is a rather additive physical parameter which is acceptable only at the first steps of the study.
  • In further studies ESEM (environmental scanning electron microscopy) instead of the conventional SEM is more suitable for the layered water-absorbing materials, since in this case the sample imaging can be performed in a chamber with a predetermined humidity instead of a deep vacuum.
  • It would be also interesting to measure local dynamic conductivity / impedance of the sandwich components due to the different localization of the electrolyte (salt) within the layered packaging material. Of particular interest is variation of the above parameters at different humidity.

Author Response

I would like to thank you for reviewing the manuscript (I revised the article). The comments and suggestions have been invaluable and I would like to address your suggestions. The research work was funded via the CORNET Programme (as the part of research project called “HumidWRAP”).  The PN-ISO 11036:1999 and PN-A-74108 standards were chosen to perform the storage tests of bread, and at the beginning of the project the methods were accepted by all partners. The suggestion that NMR relaxometry should be used for the estimation of water content, both in bread and the packaging material is very valuable, as well as using ESEM rather than SEM, or measure local dynamic conductivity. We are very grateful for these suggestions. We are going to perform these tests at the next stage of the project where it is planned to compare packaging materials received from our international partners with ‘’sandwich” packaging. Within these tests the storage tests of bread and the other bakery products will be examined.

The results of this study were obtained as a first step of the project. The tests were performed according to accepted standards, and it is not possible to carry out any additional analysis at this time because we do not have the equipment to perform NMR relaxometry or ESEM analysis, and funding for subcontracting costs were not offered (We can not pay for additional tests for the workpackage which was finished). Additionally the bread used during storage tests was purchased as one batch. We aimed to discover the differences between bread samples after storage to compare packaging materials. A preliminary study was carried out and confirmed that there were many marked differences between bread samples if the bread was purchased from different batches. This is an additional reason why we could not perform the tests according to your suggestions and introduce any new results into this article. We are mindful of your suggestion and will follow your advice during the experiments in the case of the next article.

I would like to mention that I sent the manuscript to the native speaker (according to the suggestion that moderate English changes are required). This text has been check by Simon Bretherton a native speaker from the UK.

Your sincerely

Małgorzata Mizielińska

Reviewer 2 Report

The authors reported that the sandwich form of packagings were assembled by two sheets of paper with an external Aquaseal coating and an internal starch coating (with a NaCl filler). The purposes of the internal starch/NaCl-based coating and the external Aquaseal coating were to absorb the water released by the packed bread and increase the water vapour barrier. The results show that sandwich packaging can keep the bread fresher than commercial paper packaging. Therefore, I recommend the publication in coatings. Here are some tips:

  1. If possible, it is better to compare the sandwich packaging with other packagings in different materials.
  2. If possible, please add the SEM of sandwich packaging after packing the bread for different time.
  3. Please add the specific calculation formula of how to get the statistical results.
  4. Please explain why the gumminess and springiness of bread can change as the results.
  5. If possible, please show how long will this sandwich packaging keep the bread fresh.
  6. Please explain whether there are more application scenarios for this study.

Author Response

I would like to thank you for reviewing the manuscript. The comments and suggestions have been invaluable and I would like to address your suggestions.

1. If possible, it is better to compare the sandwich packaging with other packagings in different materials.

 - Thank you for this suggestion I introduced the comparison into the discussion section.

2. If possible, please add the SEM of sandwich packaging after packing the bread for different time.

 - I added the SEM picture and the explanation to the results section.

3. Please add the specific calculation formula of how to get the statistical results.

Statistical significance was determined using an analysis of variance (2way ANOVA). All analyses were performed with GraphPad Prism 8 (GraphPad Software, San Diego, US).

4. Please explain why the gumminess and springiness of bread can change as the results. 

- Thank you for this suggestion I introduced several questions into the discussion section.

5. If possible, please show how long will this sandwich packaging keep the bread fresh.

6. Please explain whether there are more application scenarios for this study.

  • The research work was funded via the CORNET Programme (as the part of research project called “HumidWRAP”). There are industrial partners in committee useres involved in the project. The industrial partners were anxious to extend the shelf-life of bread from 2 to 3 days – which led us to examine the bread after 72 hours of storage. On obtaining positive results, the sandwich packaging and the other packaging materials obtained in this project will be used to perform storage tests on bakery products. As a result of this new data we will be writing another article. As Polish members are responsible for bakery products, we are forced solely focus on these kinds of products.

I would like to mention that I sent the manuscript to the native speaker (according to the suggestion that moderate English changes are required). This text has been check by Simon Bretherton a native speaker from the UK.

Your sincerely

Małgorzata Mizielińska

Round 2

Reviewer 2 Report

The author revised all the questions, the manuscirpt can be published on Coatings now.